# Evaluating Treatment Preferences and the Efficacy of Capsaicin 179 mg Patch vs. Pregabalin in a Randomized Trial for Postsurgical Neuropathic Pain in Breast Cancer: CAPTRANE

**DOI:** 10.3390/cancers17020313

**Published:** 2025-01-19

**Authors:** Denis Dupoiron, Florent Bienfait, Valérie Seegers, François-Xavier Piloquet, Yves-Marie Pluchon, Marie Pechard, Karima Mezaib, Gisèle Chvetzoff, Jésus Diaz, Abesse Ahmeidi, Valérie Mauriès-Saffon, Nathalie Lebrec, Sabrina Jubier-Hamon

**Affiliations:** 1Anaesthesiology and Pain Department, Institut de Cancérologie de l’Ouest, 49055 Angers, France; florent.bienfait@ico.unicancer.fr (F.B.); nathalie.lebrec@ico.unicancer.fr (N.L.); sabrina.jubier-hamon@ico.unicancer.fr (S.J.-H.); 2Biometrics Department, Institut de Cancérologie de l’Ouest, 49055 Angers, France; valerie.seegers@ico.unicancer.fr; 3Oncology and Medical Specialties Department, Institut de Cancérologie de l’Ouest, 44800 Saint-Herblain, France; francois-xavier.piloquet@ico.unicancer.fr; 4Pain Management Consultation Center, Centre Hospitalier Départemental Vendée, 85000 La Roche-sur-Yon, France; yves-marie.pluchon@ght85.fr; 5Institut Curie Hôpital de Saint-Cloud, 92210 Saint-Cloud, France; marie.pechard@curie.fr; 6Gustave Roussy Cancer Campus, 94805 Villejuif, France; karima.mezaib@gustaveroussy.fr; 7Centre Léon Bérard, 69008 Lyon, France; gisele.chvetzoff@lyon.unicancer.fr; 8Anaesthesiology and Pain Department, Institut du Cancer de Montpellier, 34090 Montpellier, France; jesus.diaz@icm.unicancer.fr; 9Department of Anesthesiology, Intensive Care, Centre Oscar Lambret, 59000 Lille, France; a-ahmeidi@o-lambert.fr; 10Department of Supportive Care, Oncopole Claudius Regaud, 31100 Toulouse, France; mauries-saffon.valerie@iuct-oncopole.fr

**Keywords:** breast cancer, capsaicin, postsurgical neuropathic pain, pregabalin

## Abstract

The CAPTRANE clinical trial explored the comparative efficacy and tolerability of high-concentration capsaicin patch (HCCP) and oral pregabalin in treating postsurgical neuropathic pain (PSNP) following breast cancer surgery. PSNP is a common and impactful complication, affecting up to 58% of patients post surgery. This multicenter, randomized trial, conducted at nine centers across France, aimed to confirm that a single HCCP treatment was noninferior to daily oral pregabalin in reducing pain scores 2 months after treatment initiation. Recruitment challenges limited participation to 140 patients (target: 644), with 107 included in the per-protocol analysis (HCCP: n = 65; pregabalin: n = 42). The change from baseline in pain scores indicated that HCCP was not inferior to pregabalin. HCCP also showed a significant reduction in painful area size, was well tolerated, and was preferred by patients. The study offers valuable data supporting HCCP as a viable alternative to pregabalin for PSNP management post breast cancer surgery.

## 1. Introduction

Breast cancer is the most common form of cancer among women in Europe and the US [1,2]. Overall survival for individuals diagnosed with breast cancer increased from 80% for women diagnosed between 1989 and 1993 to 87% for those diagnosed between 2010 and 2015 [3]. In 2018, the standardized net survival at 5 years was 87%, and at 10 years, it was 76% [3]. Therefore, patients have to adapt to living with both the disease itself and the side effects arising from treatment [4]. Neuropathic pain following breast surgery can be caused by local nerve damage, most commonly to the intercostobrachial nerve (ICBN), intraoperative damage to pathways of the axillary nerve, neuroma formation, and nerve entrapment due to scar fibrosis, with a diminished density of intraepidermal nerve fibers in mastectomy scars suggesting the presence of small fiber neuropathy [5]. Global prevalence is estimated to be between 32% and 58% [6] but is highly heterogeneous, depending on the study and pain assessment method.

Diagnosing neuropathic pain, which remains a challenge [7,8], involves questionnaires like the Douleur Neuropathique en 4 Questions (DN4), which has high sensitivity and specificity for detecting peripheral neuropathic pain (PNP) [9].

Diagnostic difficulties can delay the introduction of appropriate analgesic therapy, and chronic neuropathic pain can develop [10,11]. In the absence of a consensus on managing neuropathic pain in patients with cancer, oral treatments such as tricyclic antidepressants, antiepileptic drugs, and opioids are used. The choice of therapy depends on various factors including efficacy, side effects, cost, and concomitant medication use [12,13,14,15]. High-concentration capsaicin patch (HCCP) (179 mg QUTENZA^®^; Grünenthal, Aachen, Germany), authorized in Europe since 2009, represents an alternative approach to treating localized postoperative neuropathic pain [16]. HCCP is indicated for managing PNP in adults and can be used alone or alongside other analgesics [16]. Its major advantage lies in its very low risk of systemic side effects and a safety profile mainly characterized by transient, local application-site reactions [16].

Studies have demonstrated the effectiveness and tolerability of HCCP in patients with neuropathic pain [17,18], and at the time of writing, it is being evaluated in a phase III trial in patients with postsurgical nerve injury [19]. HCCP has shown significant benefits over pregabalin for allodynia treatment [18]. Given the significant impact of chronic neuropathic pain on patients’ lives and the importance of the early diagnosis and treatment of this disease in patients with breast cancer, the efficacy and tolerability of early intervention with either HCCP or pregabalin in postsurgical neuropathic pain (PSNP) following breast cancer surgery were investigated. A multicentric, comparative, randomized study was launched to compare HCCP with pregabalin. The primary objective of the study was to demonstrate noninferiority between HCCP and oral pregabalin regarding the change from baseline in the Numeric Pain Rating Scale (NPRS) score at month 2 in patients with neuropathic pain following surgery for breast cancer (primary endpoint).

## 2. Materials and Methods

### 2.1. Setting

CAPTRANE (CApsaïcine Précoce: évaluation dans le TRAitement des NEvralgies inter costo brachiales [NICB] post chirurgie mammaire [early capsaicin: evaluation in the treatment of ICN post breast surgery]) (ClinicalTrials.gov identifier: NCT03794388) was coordinated by the Institut de Cancérologie de l’Ouest (ICO) in France. Patients were recruited from nine centers in Angers, Villejuif, La Roche-sur-Yon, Lille, Lyon, Saint-Herblain, Saint-Cloud, Montpellier, and Toulouse, all located in France.

### 2.2. Patients

#### 2.2.1. Inclusion Criteria

In this study, patients were male or female adults (aged ≥ 18 years) who had undergone first-line surgical treatment for breast cancer, regardless of surgery type. They were required to have experienced neuropathic pain in the breast and/or axillary area corresponding to intercostobrachial neuralgia with a DN4 score of 4 or higher, identified between 3 and 12 months post surgery. Seven of the ten items in the DN4 questionnaire relate to painful and nonpainful sensory symptoms based on a patient interview, and three are based on clinical examination (hypoesthesia and allodynia). Each item has a yes (1 point) or no (0 point) response. Overall, DN4 scores ≥ 4 suggest the presence of neuropathic pain [9]. Eligible patients had to have healthy, non-irritated skin in the painful areas targeted for treatment. All patients provided written informed consent.

#### 2.2.2. Exclusion Criteria

Patients were excluded if they had specific contraindications to the study treatments, including hypersensitivity to the active substances or excipients of either capsaicin or pregabalin. Patients with diabetes were not eligible, nor were those who had previously been treated with capsaicin or pregabalin between their surgery and inclusion in the study. Ongoing opioid therapy exceeding 80 mg/day (oral morphine equivalent) at the time of inclusion, the use of topical pain treatments within 7 days of inclusion, uncontrolled hypertension (systolic blood pressure ≥ 180 mmHg or diastolic blood pressure ≥ 90 mmHg), a recent history (<3 months) of cardiovascular events (stroke, myocardial infarction, or pulmonary embolism), chronic kidney disease, pregnancy, potential pregnancy, and breastfeeding were also criteria for exclusion. Additionally, individuals legally deprived of liberty or under guardianship and those unable to undergo medical follow-up for the trial duration were not considered for participation.

### 2.3. Study Design

CAPTRANE was a phase III, multicenter, randomized, parallel-arm, open-label study undertaken to demonstrate the noninferiority of early topical treatment with HCCP 8% (QUTENZA^®^, Grünenthal, Aachen - Germany) relative to oral pregabalin after 2 months in individuals with neuropathic pain following surgery for breast cancer. The study recruited patients from expert pain centers in France from March 2019 and comprised a 6-month treatment period with HCCP or pregabalin followed by a crossover design where patients initially receiving HCCP could switch to pregabalin and vice versa (Figure 1). Recruitment was completed in May 2022 and the final follow-up visit took place in November 2022.

The study protocol was approved by the appropriate institutional review board/independent ethics committee at each participating study site and by any relevant competent authority (Ref: 18 1104). The study was conducted in compliance with the protocol and ethical principles within the Declaration of Helsinki, and in accordance with the Council for International Organizations of Medical Sciences International Ethical Guidelines and the International Conference on Harmonisation Good Clinical Practice Guideline. All patients provided written informed consent prior to their enrollment into the study.

### 2.4. Randomization

Eligible patients were randomized to receive either HCCP 8% or oral pregabalin tablets. Randomization was executed in variable balanced blocks using a minimization algorithm and Ennov software, Version 10.3 (Paris, France), and was stratified by treatment center, pain intensity on the NPRS (score < 5 or ≥5), participant age (≤65 or >65 years), and anxiety score during the screening period on the Hospital Anxiety and Depression Scale (HADS) (score < 8, 8–10, or ≥11). This stratification ensured a balanced distribution of these variables across the study arms, enhancing the reliability of the comparison between the two treatment modalities.

The NPRS is a unidimensional measure of pain intensity in adults. It is an 11-point numeric scale ranging from 0 (no pain) to 10 (worst pain imaginable) [20]. HADS is a single questionnaire, comprising fourteen questions (seven for anxiety [HADS-A] and seven for depression [HADS-D]), which can be self-administered within 2–5 min and facilitates the early identification of both anxiety and depression simultaneously [21].

### 2.5. Study Treatments

For patients randomized to receive HCCP, a maximum of two patches were applied simultaneously to the painful area for a maximum duration of 60 min. If the affected surface area was too large to be treated in a single session, a second session could be carried out within 8 days following the first session to complete this application, these two sessions being considered as a single application. The application of capsaicin patches was repeated once, 3 months after the initial application, in cases of pain persistence or recurrence.

For patients randomized to receive pregabalin, renal function was checked prior to treatment initiation. Patients were subsequently treated with a dose of 50 mg/day in two divided doses (or 25 mg/day if deemed appropriate by the treating physician) and uptitrated every 3–7 days by up to 50 mg/day to a maximum dose of 600 mg/day, as appropriate. A compulsory interval of 3 days was maintained between each dose increment. Patients were evaluated 2 months (±7 days) after treatment initiation, after which they could choose to continue with their assigned treatment or switch, provided a 3-month interval between HCCP applications was maintained.

### 2.6. Assessments

#### 2.6.1. Primary Endpoint

The primary endpoint was the difference between the two treatment groups in terms of a change in the NPRS score (range: 0–10) from baseline to month 2. The change from baseline in the highest pain intensity reported in the previous 24 h was compared. This measurement involved collecting pain intensities during various times of the day—during the night, evening, afternoon, and morning. Pain intensity was assessed at the time of randomization and again at the 2-month assessment visit.

#### 2.6.2. Secondary Endpoints

Secondary outcomes in the study were focused on month 2 (±7 days) post randomization. Assessments were conducted at baseline, at 2 months, and additionally at 6 months (this manuscript focuses on the short-term data; long-term data will be reported later). These outcomes provided insights into both the clinical effectiveness and patient-reported experiences of the treatments within the first 2 months after one HCCP treatment or daily intake of pregabalin.

##### Patient-Reported Outcomes

The Patient Global Impression of Change (PGIC) was used to gauge the patients’ overall perception of treatment effectiveness. Quality of life (QoL) was assessed using the European Organisation for Research and Treatment of Cancer (EORTC) 30-item Core Quality of Life Questionnaire (QLQ-C30) [22] for use in patients with cancer, and the EORTC 23-item Breast Cancer Quality of Life Questionnaire (QLQ-BR23) for use in patients with breast cancer specifically [23]. The EuroQol Five-Dimension 5-Level questionnaire (EQ-5D-5L) complemented these assessments by providing insights into functional and overall QoL changes.

##### Clinical Assessments

Baseline characteristic data were collected and included demographic information, cancer treatment history, and information on the surgical procedure leading to PSNP. The NPRS was used until the end of the study for a continued evaluation of pain intensity. The treatment surface area was determined through pain mapping by the treating physician, copied on tracing paper. Tracing paper was sent for central reading (Sketchandcalc) and the analysis of the painful region. Body weight measurements were recorded to monitor any significant changes over the course of the treatment.

##### Psychological and Tolerability Assessments

The HADS was employed to evaluate the psychological impact of the treatment on patients. Furthermore, the study quantified treatment tolerance through the documentation of adverse events (AEs) of grade ≥ 2, as defined by the Common Terminology Criteria for Adverse Events (CTCAE) version 5.0. Treatment tolerance was quantified through the documentation of grade ≥ 2 AEs, according to CTCAE version 5.0.

##### Exposure to Trial Treatments

In the HCCP treatment arm, information on the total number of patches received by each participant after randomization was collected. In the pregabalin treatment arm, information on the daily dose was collected.

### 2.7. Statistical Analyses

#### 2.7.1. Sample Size Calculation

Assuming a standard deviation (SD) in the distribution of NPRS scores at baseline of 2.0 and a difference in mean NPRS scores between HCCP and pregabalin of 0, an a priori noninferiority limit was set at +0.4 points on the NPRS with a unilateral alpha risk of 4.5%. No universally accepted clinically meaningful difference exists between treatment and comparator [24] and we considered that a difference less than 50% of the active comparator’s effect compared with placebo would be an acceptable margin [25]. Data from pregabalin RCTs of at least 12 weeks’ duration demonstrated mean differences between placebo and pregabalin 300–600 mg ranging from −0.97 to −1.79 points on the NPRS, depending on the study [26]. Based on this, a noninferiority margin of 0.4 points was deemed appropriate.

Based on these assumptions, it was determined that 644 patients (322 patients per treatment group) would be required to demonstrate noninferiority with a power of 80% between HCCP and pregabalin in terms of difference between treatment groups in the change in NPRS between baseline and month 2. To account for the possibility of having 20% of patients non-evaluable at the end of the 6-month treatment period, the total number of patients required for the study was adjusted to 772 (386 patients per arm). If the observations in the trial are accurate and considering the allocation ratio in the per protocol population (a posteriori estimate), the empirical power is 38%.

An interim analysis was initially planned after the inclusion of 300 patients (at least 150 patients per arm) for the primary endpoint. The allocation of the alpha risk for this was 0.005 for the interim analysis and 0.045 for the final analysis. However, the interim analysis was not conducted due to recruitment challenges, leading to a consolidation of the alpha risk at 5% for the final analysis.

#### 2.7.2. Analyses Conducted

The primary endpoint at month 2 was analyzed in the per-protocol (PP) population, comprising all patients who were randomized and received the treatment corresponding to their randomization arm; patients assigned to pregabalin were required to have received at least 75% of the prescribed dose. The causal effect of the treatment itself was the primary focus of this research. Accordingly, the PP population was selected for the analysis of the primary endpoint. Patients who received a treatment other than the allocated one were excluded for this primary analysis. For the primary analysis, only patients with NPRS assessments at baseline and month 2 were considered and no imputation for missing data was performed.

Additionally, two analyses were performed on the PP analysis set. The predefined sensitivity analysis accounted for missing NPRS data as follows: if the randomization score was missing, the screening value was used for both groups. For missing month 2 scores, the score at inclusion was used in the HCCP group whereas in the pregabalin group, a score of 0 (no pain) was used. As this approach severely penalized HCCP in cases of missing data, an additional sensitivity analysis was defined by imputing missing data at month 2 using the overall NPRS score change across the entire cohort.

In line with the International Council for Harmonisation of Technical Requirements for Pharmaceuticals for Human Use E9 guidelines, the confidence interval (CI) approach has a one-sided hypothesis test counterpart for testing the null hypothesis that the treatment difference (investigational product minus control) is equal to the lower equivalence margin versus the alternative that the treatment difference is greater than the lower equivalence margin. The 90% CI of the mean change in NPRS score in the HCCP group was estimated using a bias correction and accelerated percentile bootstrap resampling method, due to the non-Gaussian distribution. To establish noninferiority and reject the null hypothesis, the upper bound of the 90% CI for the mean change in NPRS score in the HCCP group at month 2 had to be less than 0.4 points (clinically predefined margin) higher than the mean change in NPRS score in the pregabalin group.

All secondary endpoints were analyzed using data from the modified intention-to-treat (mITT) population, which included all randomized patients who received at least one administration of the treatment. This analysis, evaluating the first 2 months after first treatment, included patients according to the treatment arm they were allocated to, regardless of the actual treatment received. Patients who were prematurely withdrawn or wrongly included before receiving any treatment were not included in this analysis. No imputation for missing data was undertaken. Quantitative variables are summarized using mean, SD, median, and the 25th and 75th percentiles. Results between the two treatment groups were compared with an analysis of variance (ANOVA) when residuals followed a normal distribution. Otherwise, non-parametric tests were used including the Mann–Whitney test for independent data and the Wilcoxon test for paired data. The alpha risk was not corrected for multiple comparisons. Categorical and binary variables were summarized using number and percentage. All analyses were conducted in R (R Core Team, 2023), version 4.1.2.

## 3. Results

### 3.1. Patients

The study faced challenges with participant recruitment, partially due to the coronavirus disease 2019 (COVID-19 pandemic), which led to the trial being halted earlier than anticipated. Of the 437 patients who provided informed consent for the study, a total of 140 met the criteria for randomization (Figure 2). The mITT population comprised 116 patients (HCCP: n = 65; pregabalin: n = 51) and the PP population comprised 107 patients (HCCP: n = 65; pregabalin: n = 42).

An equal number of 70 patients were allocated to each study arm. However, treatment initiation differed slightly between the groups: 65 patients received HCCP treatment, all of whom adhered to the protocol; in the pregabalin arm, only 51 patients started treatment, 42 of whom followed the protocol specifications. At the conclusion of the initial 2-month period of the trial, 46 patients in the HCCP group opted to continue with HCCP, and no participant expressed a preference to switch to the pregabalin group. In contrast, 19 patients within the pregabalin group chose to continue with their assigned treatment, while 27 expressed a preference to switch to HCCP. Among the 27 patients randomized to the pregabalin arm who expressed a preference to switch to HCCP, 6 patients (22.2%) received HCCP before reaching the 2-month endpoint.

### 3.2. Participant Demographics and Exposure to Treatment

Participant demographics and disease characteristics are shown in Table 1. The treatment groups were well balanced, with no significant differences between them. All patients were female, predominantly under the age of 65 years, with most undergoing mastectomy, lumpectomy, or zonectomy. The surgical procedure distribution varied across treatment arms; lumpectomy was slightly more common in the pregabalin arm (37.3%), while zonectomy was more frequent in the HCCP arm (40.0%). A higher proportion of patients in the HCCP group (73.8%) received hormonal therapy compared with the pregabalin arm (64.7%). Lymphoedema was the most frequently reported complication of breast surgery. The NPRS scores at baseline varied between 3 (min) and 10 (max).

At month 2, patients randomized to HCCP received a mean of 1.0 patches (range: 0.2–4.0) at a median time between randomization and first HCCP application of 12 days (range: 0–91 days) (PP and mITT populations). In the pregabalin arm, 94.1% of patients in the mITT population (n = 51) and 100% in the PP population (n = 42) had received ≥ 75% of the prescribed doses by month 2. The median maximum dose was 150 mg (range: 25–300 mg) per day (mITT and PP populations).

### 3.3. Primary Endpoint (Change in NPRS Scores)

Baseline NPRS scores were similar between treatment groups. The primary endpoint of a change from baseline at 2 months in NPRS score within the PP population (non-imputed) showed a mean (SD) reduction of −1.926 (2.554) in the HCCP group (n = 54) and −1.634 (2.498) in the pregabalin group (n = 41) (Figure 3). Against a noninferiority margin of +0.4, for patients receiving HCCP, the observed 90% CI between-arm difference change was −0.889 to +0.260. Therefore, the null hypothesis was rejected in this context, indicating that HCCP treatment is noninferior to pregabalin in reducing NPRS score at the margin of 0.4, as the change observed did not fall below the noninferiority threshold when compared with pregabalin.

Two sensitivity analyses were conducted on the total PP population of patients randomized to HCCP patients (n = 65) or pregabalin (n = 42) to account for missing data. For the primary sensitivity analysis, missing NPRS data in the HCCP group were imputed to the baseline value whereas for the pregabalin group, missing data were set to 0 (i.e., complete disappearance of pain at month 2). This strict preset strategy to account for missing data severely penalized HCCP as seven patients in the HCCP group had missing data compared to one in the pregabalin group. The 90% CI was −0.417 to +0.552, leading to the decision not to reject the null hypothesis. Further evaluation of the change from baseline scores for each patient considered in the PP analyses, however, shows a similar pattern across treatments (Figure 4).

An additional post hoc sensitivity analysis taking the mean change from baseline for the total study population was used to impute missing NPRS data. The 90% CI was −0.743 to +0.192, leading to the rejection of the null hypothesis.

### 3.4. Functional Change (According to PGIC Scores)

At month 2, the majority of patients in both groups reported some degree of improvement in functional change as measured by the PGIC. In the HCCP group, 68.7% of patients experienced some level of positive change, including 27.5% of patients who reported feeling “a little better”, 21.6% noting that they were “somewhat better but the change does not make much difference”, and 19.6% experiencing a “moderate but noticeable” improvement. In the pregabalin group, 68.9% of patients reported some improvement, distributed as 34.5% feeling “a little better”, 24.1% feeling “somewhat better”, and 10.3% noting a “moderate but noticeable” improvement.

It is important to note that the PGIC results had a high rate of missing responses, especially in the pregabalin arm (43.1%) compared with the HCCP group (21.5%). No change in condition or a worsening of symptoms was reported in 23.5% of patients in the HCCP group and 20.7% in the pregabalin group.

### 3.5. QoL (Measured Using EQ-5D-5L, EORTC QLQ-C30, and EORTC QLQ-BR23)

The EQ-5D-5L utility scores in the HCCP group improved from a mean (SD) score of 0.635 (0.236) at baseline to 0.718 (0.214) at 2 months (*p* = 0.043); in the pregabalin group, mean (SD) scores increased from 0.596 (0.23) to 0.652 (0.284) (*p* = 0.29). These results indicate an overall enhancement in perceived health-related QoL for HCCP. The observed changes were not statistically significant when comparing the two treatment arms (*p* = 0.18; Student’s *t* test). After 2 months, HCCP showed important benefits in terms of cognitive capacity (*p* = 0.006) and reduction in nausea/vomiting symptoms (*p* = 0.023), whereas pregabalin notably enhanced work/leisure capacity (*p* = 0.004) and future outlook (*p* < 0.001).

There were no differences in the results of QLQ-C30 or QLQ-BR23 scores at baseline between the treatment groups. However, at month 2, there was a statistically significant difference for the constipation dimension (*p* = 0.0223; cross-sectional analysis, *p*-value calculated using ANOVA test): a decrease was noted in the HCCP arm and an increase in the pregabalin arm. The QLQ-C30 and QLC-BR23 assessments demonstrated significant improvements across several dimensions. The mean QoL score (SD) in the HCCP group increased from 56.2 (19.7) at baseline to 61.3 (20.0) at month 2 (*p* = 0.18), compared with stable scores in the pregabalin group from 56.4 (19.4) to 56.3 (22.1) (*p* = 0.91).

Within the HCCP group, significant improvements over time were observed in several QoL dimensions (*p*-value calculated with Wilcoxon’s test for paired data in participants who contributed data at the two measurement times), including physical functional capacity (*p* = 0.005), fatigue (*p* < 0.001), pain symptoms (*p* < 0.001), dyspnea symptoms (*p* = 0.044), constipation symptoms (*p* = 0.031), and breast symptoms (*p* < 0.001). In the pregabalin group, significant improvements were noted. These encompassed pain symptoms (*p* = 0.025), constipation symptoms (*p* = 0.009), and breast symptoms (*p* = 0.001).

### 3.6. Reduction in Painful Area

At baseline, the mean (SD) painful area was 115.3 cm^2^ (79 cm^2^) for the HCCP group and 130.8 cm^2^ (79.5 cm^2^) for the pregabalin group. The median (range) values were 118.6 cm^2^ (3.6–309.9 cm^2^) for HCCP and 127.7 cm^2^ (7.1–337.1 cm^2^) for pregabalin. The difference between the groups at this stage was not statistically significant (*p* = 0.3022 for mean values and *p* = 0.3083 for median values).

At the 2-month follow-up, a significant reduction in the painful area was observed in both groups. The mean (SD) painful area was reduced to 66.1 cm^2^ (49.9 cm^2^) in the HCCP group, compared with 91.9 cm^2^ (63.3 cm^2^) in the pregabalin group. The median (range) painful area also showed a similar trend, with a reduction to 56.6 cm^2^ (0–204.6 cm^2^) in the HCCP group and to 80.7 cm^2^ (0–224.1 cm^2^) in the pregabalin group. The differences between the groups at this point were statistically significantly favoring HCCP (*p* = 0.0205 for mean values and *p* = 0.0471 for median values).

### 3.7. Anxiety and Depression (Measured Using HADS)

At baseline, the mean (SD) anxiety scores as measured by HADS were 8.5 (4.0) for the HCCP group and 8.8 (3.2) for pregabalin. By month 2, the mean (SD) anxiety scores decreased to 7.8 (4.1) and 8.2 (3.4), respectively.

Regarding depression, the mean (SD) scores at baseline were 6.4 (4.2) for HCCP and 6.1 (3.7) for pregabalin. By month 2, the mean (SD) depression scores were 6.2 (4.4) and 6.2 (4.7), respectively, indicating stability in the levels of depressive symptoms in the first 2 months after treatment initiation.

### 3.8. Tolerability

No serious AEs were reported. Up to month 2, a total of 20 AEs attributable to the assigned treatments were recorded: 8 in the HCCP group and 12 in patients who received pregabalin. Up to month 2, the following AEs were recorded for the HCCP group: one event of a burning sensation in one participant, and seven events of a burning sensation at the site of application associated with pain in seven patients. In the pregabalin group, up to month 2 and before switching from pregabalin to HCCP, there were four events of vertigo in four patients; six events of ataxia, somnolence, imbalance, memory problems, and difficulty concentrating in five patients; one event of diarrhea in one participant; and one event of asthenia in one participant.

## 4. Discussion

This is the first trial to explore the efficacy and tolerability of a single topical treatment compared with a daily oral treatment for PNP following breast cancer surgery over a period of 2 months. The population included in this trial consisting solely of female patients (the majority aged < 65 years) is consistent with data reported in the literature [6]. More than half of the patients underwent surgical treatment with either lumpectomy or zonectomy. Nearly 80% were treated with radiotherapy and approximately 70% underwent hormone therapy.

The primary goal of this study was to demonstrate the noninferiority of HCCP compared with pregabalin within the first 2 months following treatment initiation. The results showed noninferiority in the predefined PP analysis set. The predefined sensitivity analysis, which heavily penalized the HCCP treatment group (using a baseline observation carried forward approach) compared with the pregabalin group (assigned no pain at month 2), did not show noninferiority. A less conservative sensitivity analysis defined post hoc did show noninferiority. These results need to be interpreted with caution considering the reduced sample size that decreases the chance of finding a difference in treatment effect between two treatment arms. Nevertheless, the data are interesting and align with a previous, well-powered study of 629 patients that demonstrated the noninferiority of HCCP vs. pregabalin [17].

In both treatment groups, a significant decrease in NPRS scores was observed, consistent with the recognized efficacy of both treatment modalities. Between baseline and month 2, a statistically significantly greater reduction in the size of the painful area was observed in the HCCP group compared with the pregabalin group. A greater reduction in the area of allodynia favoring capsaicin vs. pregabalin has also been reported [18].

QoL, a crucial aspect of cancer survivorship, improved with both treatments. After 2 months, HCCP showed important benefits in terms of cognitive capacity (*p* = 0.006) and a reduction in nausea/vomiting symptoms (*p* = 0.023), whereas pregabalin notably enhanced work/leisure capacity (*p* = 0.004) and future outlook (*p* < 0.001). These differences in QoL improvements indicate a variable impact of the two treatments on different aspects of patient well-being. This differential impact of HCCP compared with pregabalin was also noted in another comparative study regarding patients’ perception of cognitive functioning (measured by the Medical Outcomes Study Cognitive Functioning scale) [27].

Although both treatments led to a significant decrease in NPRS scores at month 2, the dropout rate was more pronounced in the pregabalin group than in the HCCP group. This observation aligns with another comparative study in patients with PNP of various origins, in which the dropout rate in the pregabalin arm (41 of 277 patients) was much higher than in the HCCP arm (6 of 286 patients) [17]. Moreover, an analysis of the time to switch from pregabalin to HCCP revealed that 20% of patients had already switched to HCCP prior to the predefined 2-month assessment point, against the provisions of the protocol. This may be attributed to the titration required for pregabalin treatment, which may have caused a delay in the onset of efficacy or may be due to adverse drug reactions associated with the uptitration of pregabalin, as has been reported previously [17]. Data on the subsequent phase of the trial covering the remaining 4 months of the study will be presented separately.

Both treatments were generally well tolerated, with HCCP mostly causing application-site reactions and pregabalin most frequently leading to systemic AEs. Notably, a previous study found a lower burden of treatment with HCCP vs. pregabalin [28]. More patients in the HCCP group continued with this treatment, and none switched to pregabalin, whereas 27 of 51 patients switched from pregabalin to HCCP. The number of patients wanting to switch from oral to topical treatment is noteworthy.

Schubert et al. noted that key factors influencing the choice of treatment were the onset of effect (odds ratio [OR]: 2.141 [95% CI, 1.837–2.494]), followed by the risk of systemic side effects (OR: 2.038 [95% CI, 1.731–2.400]) and the risk of sexual dysfunction (OR: 1.839 [95% CI, 1.580–2.140]), with the risk of local skin side effects being less of a concern (OR: 1.612 [95% CI, 1.321–1.966]) [29]. Of the tested baseline characteristics, only anxiety seemed to impact treatment outcomes significantly. Other factors tested, such as the duration of PSNP and the presence of depression, were not found to be impactful.

The trial faced challenges, including a smaller-than-planned sample size caused by the advent of the COVID-19 pandemic, which also contributed to a high dropout rate and a significant amount of missing data by limiting visits to trial sites. Despite efforts to reach the target enrollment, the final number of patients was lower than planned, largely due to the significant impact of the COVID-19 pandemic on the recruitment of clinical trials [30]. Further limitations include the open-label nature of the study and the definition of the primary assessment at month 2 after one treatment with HCCP only (i.e., not including minimally three treatment cycles as currently recommended in the EU SmPC). The trial limitations should be considered when interpreting these data.

## 5. Conclusions

This study marks the first investigation into the efficacy of HCCP in patients with cancer experiencing PSNP early after breast surgery. The primary endpoint, which focused on pain intensity reduction as measured by the NPRS, showed important reductions in pain intensity with both treatments and indicates that HCCP treatment is noninferior to pregabalin in reducing NPRS scores at the 0.4 margin in the PP analysis after 2 months. Another notable finding was the reduction in the size of the painful area, particularly in the HCCP group, within the first 2 months after treatment initiation.

Importantly, the study also revealed a distinct patient preference: while several patients switched from pregabalin to HCCP, none switched from HCCP to pregabalin. Despite its limitations, including the short follow-up period after treatment initiation, this study contributes valuable initial data to the field of PSNP management in breast cancer care.

## Figures and Tables

**Figure 1 cancers-17-00313-f001:**
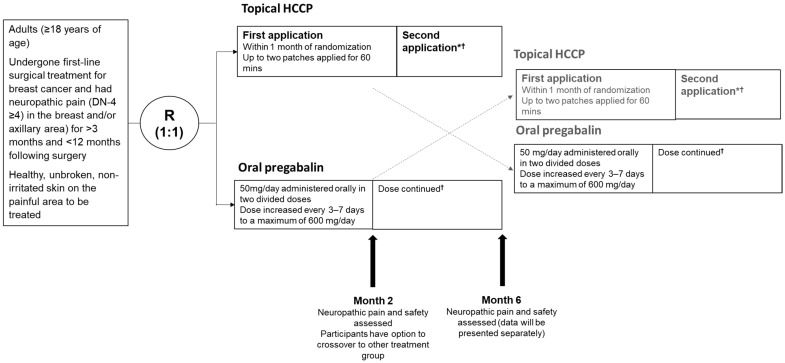
Study design. * A second application of HCCP can only be applied after a minimum of 3 months. † The continuation of study treatment is based on its tolerability and whether the neuropathic pain persists. DN4, Douleur Neuropathique en 4 Questions; HCCP, high-concentration capsaicin patch; R, randomization.

**Figure 2 cancers-17-00313-f002:**
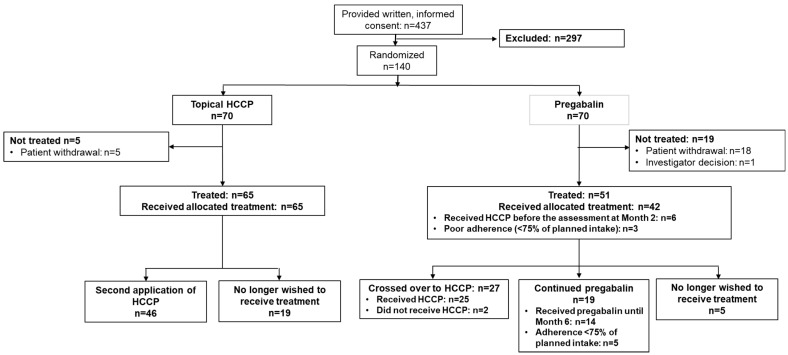
Participant flow.

**Figure 3 cancers-17-00313-f003:**
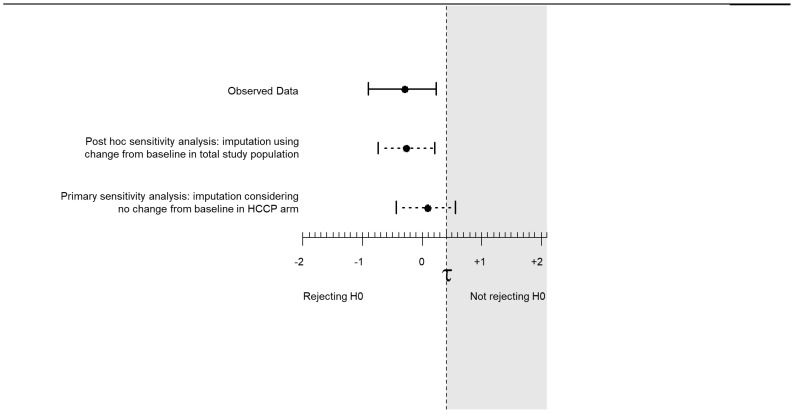
Primary endpoint: change from baseline for the two treatment groups at 2 months in NPRS score (PP population). H0, null hypothesis; HCCP, high-concentration capsaicin patch; NPRS, Numeric Pain Rating Scale; PP, per-protocol.

**Figure 4 cancers-17-00313-f004:**
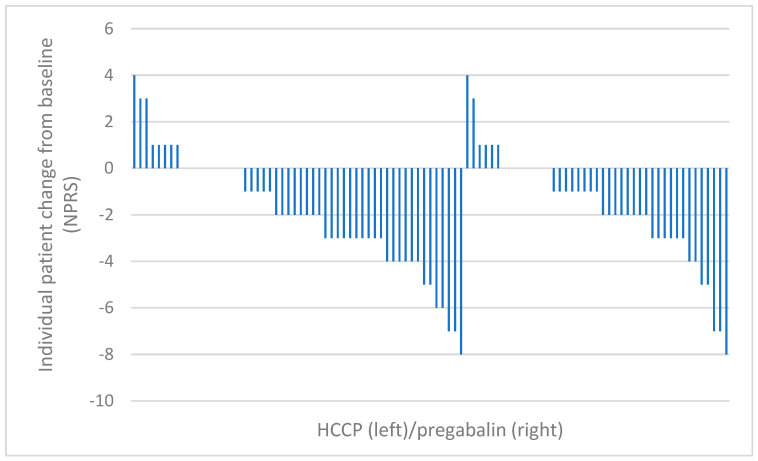
Change from baseline in NPRS score at month 2 after treatment initiation for each individual patient in the per-protocol analysis. HCCP, high-concentration capsaicin patch; NPRS, Numeric Pain Rating Scale.

**Table 1 cancers-17-00313-t001:** Participant baseline demographics and disease characteristics (PP population).

Characteristic	HCCP (n = 65)	Pregabalin (n = 42)
Age, years, median (range)	55.0 (30–86)	57.5 (34–72)
<65 years	49 (75.4)	31 (73.8)
≥65 years	16 (24.6)	11 (26.2)
Female	65 (100)	42 (100)
NPRS score at randomization	6.1 (1.5)	6.3 (1.7)
Type of primary breast surgery		
Mastectomy	18 (27.7)	11 (26.2)
Lumpectomy	19 (29.2)	17 (40.5)
Zonectomy	26 (40.0)	14 (33.3)
Not specified	2 (3.1)	0
Concomitant surgery		
Sentinel lymph node biopsy	41 (63.1)	28 (66.7)
Axillary dissection	21 (32.3)	14 (33.3)
Immediate reconstruction	16 (24.6)	7 (16.7)
Anesthesia		
General	65 (100)	42 (100)
Paravertebral block	1 (1.5)	0
Interpectoral block	2 (3.1)	0
Surgical complications		
Hematoma	8 (12.3)	4 (9.5)
Infection	2 (3.1)	0
Lymphoedema	15 (23.1)	9 (21.4)
Cancer treatment		
Radiotherapy	52 (80.0)	36 (85.7)
Hormonotherapy	48 (73.8)	29 (69.0)
Neoadjuvant chemotherapy	15 (23.1)	6 (14.3)
Adjuvant chemotherapy	19 (29.2)	12 (28.6)

Data are n (%), unless stated otherwise. NPRS, Numeric Pain Rating Scale; PP, per-protocol. There were no statistically significant differences between treatment groups.

## Data Availability

The data presented in this study are available upon request from the research department of the Institut de Cancérologie de l’Ouest (ICO) due to data privacy reasons.

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
