# Peer review of "Evaluating Treatment Preferences and the Efficacy of Capsaicin 179 mg Patch vs. Pregabalin in a Randomized Trial for Postsurgical Neuropathic Pain in Breast Cancer: CAPTRANE"

_cancers, 2025, doi:10.3390/cancers17020313_

Round 1
Reviewer 1 Report
Comments and Suggestions for Authors
In my opinion the paper is well written also from a methodological point of view.
I only have two revisions:
Major (this I believe is a very delicate point which must be carefully clarified): Provide an adequate justification of the number actually used in the study which is much lower than the one estimated and clarify better the reason for a unilateral alpha error in the calculation of the sample size .
Minor: For completeness I propose to insert an evaluation of any significance of the differences in Table 1
Author Response
In my opinion the paper is well written also from a methodological point of view.
We thank the reviewer for the encouraging comment.
I only have two revisions:
Major (this I believe is a very delicate point which must be carefully clarified): Provide an adequate justification of the number actually used in the study which is much lower than the one estimated and clarify better the reason for a unilateral alpha error in the calculation of the sample size.
- Part 1 of the question pertaining to the adequate justification of the number used in the study is not entirely clear to us. Therefore, we have addressed it in two potential ways based on our interpretation of the reviewer’s concern:
Justification for the number of subjects randomized in the study
- The actual number of participants in the study was determined by the number of subjects randomized. Recruitment challenges led to a significantly lower number of participants compared with the originally estimated sample size. After three years of recruitment, it became evident that, at the current recruitment rate, achieving the planned sample size would require a further 10 years. This was not feasible due to funding constraints and a declining interest from participating sites. The COVID-19 pandemic initially caused a sharp decline in recruitment, and this rate did not recover even after the pandemic subsided.
The recruitment challenges are mentioned in the final sentence of section 2.7.1 (Sample size calculation) and the first sentence of section 3.1 (Patients). They are discussed further in the final paragraph of the section 4 (Discussion), where the implications of the reduced sample size on the study outcomes are also highlighted. Despite these limitations, we believe the results remain robust and meaningful within the context of this reduced population.
Justification for the analysis population
- If the reviewer is referring to the analysis population, it is important to clarify our approach. In superiority trials, the intent-to-treat (ITT) analysis is commonly employed to measure the effect of allocating a treatment on participant outcomes. This approach is particularly relevant when the primary research interest lies in the causal effect of treatment allocation. However, our study's primary interest was in assessing the causal effect of the treatment itself. Our research question focused on the non-inferiority of capsaicin rather than its superiority, and a per-protocol (PP) analysis is more appropriate. This approach ensures that we accurately assess the treatment’s true efficacy without the confounding effects of non-adherence.
We acknowledge that this choice may introduce potential biases, as adherent participants might systematically differ in underlying prognostic factors compared with non-adherent participants, and the PP populations in the treatment groups could differ in terms of prognostic characteristics. To address these concerns, we conducted two sensitivity analyses to ensure the robustness of our findings.
To address this concern of the reviewer, we have added the following to section 2.7.2 (Analyses conducted): “The causal effect of the treatment itself was the primary focus of this research. Accordingly, the PP population was selected for analysis of the primary endpoint.“
- To address part 2 of the question, we will clarify the rationale for using a one-sided alpha in this non-inferiority trial.
Justification for using a one-sided alpha in non-inferiority trials
Using a one-sided alpha in non-inferiority trials is recommended because the primary objective is to demonstrate that the new treatment is not significantly worse than the control treatment by more than a pre-specified margin. This approach is more suitable for non-inferiority trials as it focuses on detecting a difference in a specific direction (i.e., ensuring the new treatment is not inferior).
Our methodology aligns with the International Council for Harmonisation of Technical Requirements for Pharmaceuticals for Human Use (ICH) E9 guidelines, which provide globally recognized standards for statistical principles in clinical trials. Specifically, section 3.3.2 of these guidelines explains that one-sided tests are suitable for non-inferiority trials because they are designed to test whether the new treatment is not worse than the control by a specified margin, rather than testing for differences in both directions, as would be the case with two-sided tests.
Consequently, a one-sided approach was used to calculate the required sample size for this trial.
We refer to the ICH E9 guidelines, which are internationally recognized standards for the statistical principles applied in clinical trials. These guidelines are developed by the ICH, which brings together regulatory authorities and the pharmaceutical industry to discuss scientific and technical aspects of drug registration. In the ICH E9 guidelines, this recommendation is found in Section 3.3.2. The guidelines explain that one-sided tests are suitable for non-inferiority trials because they are designed to test whether the new treatment is not worse than the control by a specified margin, rather than testing for differences in both directions, which would be the case with two-sided tests.
For clarity, we have added the following additional text to the statistical analysis section: “In line with the International Council for Harmonisation of Technical Requirements for Pharmaceuticals for Human Use E9 guidelines, the confidence interval approach has a one-sided hypothesis test counterpart for testing the null hypothesis that the treatment difference (investigational product minus control) is equal to the lower equivalence margin versus the alternative that the treatment difference is greater than the lower equivalence margin.”
Minor: For completeness I propose to insert an evaluation of any significance of the differences in Table 1
Minor: Evaluation of significance in Table 1
Table 1 presents the baseline characteristics of the study population. P-values have been calculated, but none were found to be statistically significant. Differences of interest between groups are described in the text above the table. To address this comment, we have added the following footnote to Table 1 as follows: “There were no statistically significant differences between treatment groups.”

Reviewer 2 Report
Comments and Suggestions for Authors
This paper describes a well conducted and methodologically sound, open label investigation to examine the non-inferiority of high-concentration capsaicin patch (HCCP) compared to pregabalin. The study potentially adds to the literature since few of the data for HCCP comes from any type of randomised trial. The clinical area of persistent post-surgical breast pain is also a very important problem in the so called ‘cancer survivor’ population.
There is large short fall in recruited patients given the number required for non-inferiority as stated in the methods. 322 were required per group but only 65 and 51 were treated in the HCCP and pregabalin groups respectively.
Presenting the outcomes at 2 months is quite an early outcome time and, as has been argued for other long term neuropathic states, is of uncertain clinical significance. Why is the promised 6 month data not appearing in same paper? Given the limitations of short term data, the 6 month data is far more relevant and valid.
There are no details of how the areas to apply the patches were mapped and then subsequently measured to reflect the areas as reported.
Why were the primary outcomes done ‘per protocol’ (which has been associated with increased risk of bias)? And why were the secondary outcomes analysed with intention to treat? For the HCCP groups 65 were treated but only 54 analysed, and for the pregabalin groups 51 treated, 41 analysed.
Why was 0.4 points taken as the clinically predefined margin?
Arguably, some of the results of some of the secondary outcomes are somewhat overstated.
Minor points
Line 60: It should be noted that ICN is only evident in a relatively small number of patients with persistent post surgical breast pain.
Line 63/64 The nomenclature of ‘post mastectomy pain syndrome’ is outdated since any breast surgery can cause the persistent pain. Indeed, most of the patients in the paper did not have a mastectomy but a lumpectomy. Approximately 1/3 patients with persistent post surgical breast pain do not have evidence of neuropathic pain and so it should be stated that this study is not necessarily applicable to all patients with persistent post-surgical breast pain.
Line 75 Ref 14 Only 9% had neuropathic pain with AI
Line 77 Is a treatment from 2009 ‘novel’?
Line 110 Does not a positive DN4 ‘suggest’ neuropathic pain rather than ‘indicate’
Why is diabetes an exclusion criterium?
Line 118 Why are patients on higher doses opioid not eligible?
Was there a minimum NPRS for inclusion or would an NPRS of 1 be sufficient?
Line 205 No detail is given of pain mapping. Was a stimulus used to map evoked pain, or was it patient indicated? If stimulus, which stimulus?
Line 291 What is a zonectomy?
Line 299 If the maximum was two patches (line162), how can the range at 2 months be up to 4? (Not explainable by second application of 2 patches since second patch had to be 3 months after the first)
Line 334: Would the PGIC results be better expressed in a table? It is unclear if the authors took “somewhat better but the change does not make much difference” to be a ‘positive change’ or not.
Line 347 For EQ5D5L, were the changes statistically different to their own baselines?
Line 355 Why were differences looked for in the different QLQ measures? This microanalysis is of dubious clinical significance.
Line 359 QOL has no statistics mentioned.
Line 362 Similarly the search for differences in the QOL dimensions seems a stretch to find significant results.
Line 369 Why are mean and medians both presented?
Line 382“By month 2, the mean (SD) anxiety scores decreased to 7.8 (4.1) and 8.2 (3.4), respectively, suggesting a gradual reduction in anxiety levels over the course of treatment”. This is overstating the data and no statistics are presented.
This is a difference of <1 point which corresponds to only one slightly different response to one of the 6 (graded 0-3) questions not really consistent with a ‘reduction in anxiety levels’.
Line 391 are there any statics associated with the tolerability data?
Line 425 Why are these results being stated in the discussion and not the results section?
P=0.05 is not significant as it is not <0.05 which is the convention.
Author Response
This paper describes a well conducted and methodologically sound, open label investigation to examine the non-inferiority of high-concentration capsaicin patch (HCCP) compared to pregabalin. The study potentially adds to the literature since few of the data for HCCP comes from any type of randomized trial. The clinical area of persistent post-surgical breast pain is also a very important problem in the so called ‘cancer survivor’ population.
We thank the reviewer for the thorough assessment, constructive comments, and appreciation of the clinical relevance of the data.
There is large short fall in recruited patients given the number required for non-inferiority as stated in the methods. 322 were required per group but only 65 and 51 were treated in the HCCP and pregabalin groups respectively.
The reviewer is correct. We acknowledge the shortfall in recruited patients and recognize the limitations this imposes, which are discussed further in our response to Reviewer 1 and in section 4 (Discussion) of the manuscript. Nonetheless, we hope this publication, despite its limitations, will encourage further research in this area of unmet need.
Presenting the outcomes at 2 months is quite an early outcome time and, as has been argued for other long term neuropathic states, is of uncertain clinical significance. Why is the promised 6 month data not appearing in same paper? Given the limitations of short term data, the 6 month data is far more relevant and valid.
Justification for the 2-month endpoint
We thank the reviewer for this comment regarding the duration of treatment. We appreciate that neuropathic pain guidelines for the development of new treatments for peripheral neuropathic pain (PNP) often recommend a follow-up over 12 weeks. However, neither the capsaicin 179 mg cutaneous patch nor pregabalin are novel treatments as their efficacy in this indication has already been established. Furthermore, randomized controlled trials (RCTs) comparing these treatments with placebo have shown minimal differences between measurement points at 8 and 12 weeks.
Pregabalin data: For pregabalin a comparison between weeks 8 and 12 in patients with PNP treated with either placebo,150–600 mg, or 600 mg showed significant decreases from baseline over placebo as early as week 1 for the fixed-dose group (600 mg) and week 2 for the flexible-dose group with continued superiority over the remainder of the trial. Importantly, Figure 3 in the manuscript shows that pain relief remains steady between week 8 and week 12, with minimal additional improvement during this interval.
- Reference: Freynhagen, Rainera, Strojek, Krzysztofb; Griesing, Teresac; Whalen, Edc; Balkenohl, Michaeld. Efficacy of pregabalin in neuropathic pain evaluated in a 12-week, randomised, double-blind, multicentre, placebo-controlled trial of flexible- and fixed-dose regimens. Pain 115(3):p 254-263, June 2005).
Capsaicin data: For the capsaicin 179 mg cutaneous patch, the treatment effect increased slightly between week 8 and week 12 with an average relative change from baseline of −6.6 vs −7.1 for weeks 8 and 12, respectively
- Reference: Simpson DM, Robinson-Papp J, Van J, Stoker M, Jacobs H, Snijder RJ, Schregardus DS, Long SK, Lambourg B, Katz N. Capsaicin 8% patch in painful diabetic peripheral neuropathy: A randomized, double-blind, placebo-controlled study. J Pain. 2017 Jan;18(1):42-53).
Based on these findings, we consider week 8 to be a relevant endpoint for assessing potential treatment differences.
After the first 2 months, the trial included a crossover phase, allowing patients to switch treatments. For the readers of Cancers, we believed it was important to first focus on the data from the initial part of the trial. As the reviewer pointed out, the trial is complex, and adding more data may overwhelm the reader. As stated in section 2.6.2 (Secondary endpoints), the 6-month data is beyond the scope of this manuscript and will be presented separately.
There are no details of how the areas to apply the patches were mapped and then subsequently measured to reflect the areas as reported.
The treatment area was determined by the treating physician according to the product label. This area was clearly marked. The marked area was then exactly copied on tracing paper. Paper maps were stored in the CRF and sent to the central reading center at study end where all paper maps were scanned using a single scanning device. Subsequently the PDF files were read using Sketchandcalc (include reference https://www.sketchandcalc.com/). Outputs were subsequently checked for accuracy by one of the investigators and then analyzed.
The existing sentence in section 2.6.2.2 (Clinical assessments) has been updated as follows: The treatment surface area was determined through pain mapping by the treating physician, copied on tracing paper. Tracing paper was sent for central reading (Sketchandcalc) and analysis of the painful region. Body weight measurements were recorded to monitor any significant changes over the course of the treatment.
Why were the primary outcomes done ‘per protocol’ (which has been associated with increased risk of bias)? And why were the secondary outcomes analysed with intention to treat? For the HCCP groups 65 were treated but only 54 analysed, and for the pregabalin groups 51 treated, 41 analysed.
Justification for the choice of analysis populations
The choice of the analysis population was made based on the research question posed for each of the study’s objectives.
- Primary outcome (non-inferiority): For the primary objective, the aim was to demonstrate non-inferiority (i.e., ensuring the experimental treatment is not worse than the reference treatment by a margin of 0.4).
- Secondary outcome (superiority): For the secondary objectives, the goal was to explore differences between the two arms, applying a superiority framework.
Since our research question focused on the non-inferiority of capsaicin rather than its superiority, a PP analysis is more appropriate. This approach ensures that we accurately assess the treatment’s true efficacy without the confounding effects of non-adherence.Indeed, the bias introduced by a PP analysis tends to favor the experimental arm at the expense of the control arm. Conversely, in non-inferiority trials, using an ITT analysis may penalize the experimental treatment, as it risks diluting the treatment effect and making the two arms appear more similar, potentially leading to incorrect conclusions. For this reason, the PP analysis is often preferred for non-inferiority trials to provide a clearer estimate of treatment efficacy.
Our methodology aligns with the ICH E9 guidelines, which provide globally recognized standards for statistical principles in clinical trials. Specifically, section 3.3.2 clarifies that for non-inferiority trials, it is especially important to minimize the incidence of violations of the entry criteria, non-compliance, withdrawals, losses to follow-up, missing data, and other deviations from the protocol, and also to minimize their impact on the subsequent analyses.
We therefore consider that a PP analysis is best suited to conduct the primary analysis as this approach includes only patients who adhered to the study protocol, allowing for a clearer assessment of the treatment’s efficacy by minimizing the confounding effects of protocol deviations. The rationale for using PP analysis in non-inferiority trials is that it provides a more accurate estimate of the treatment effect by excluding non-adherent patients. This reduces the risk of bias introduced when patients who deviate from the protocol are included in the analysis. In contrast, including such patients in an ITT analysis can dilute the observed treatment effect, making the two arms appear more similar than they are and potentially leading to incorrect conclusions about non-inferiority. Additionally, the ICH E9(R1) addendum emphasizes that PP analysis is critical in non-inferiority trials as it better reflects the true performance of the treatment under ideal conditions. By focusing on patients who adhered to the protocol, the PP analysis ensures that the results accurately reflect the treatment’s efficacy, providing a more reliable basis for decision-making.
Why was 0.4 points taken as the clinically predefined margin?
Justification for the non-inferiority margin of 0.4 points
For defining the non-inferiority criteria, a pragmatic approach was adopted, as no universally accepted clinically meaningful difference exists between treatment and comparator.
- Smith SM, Dworkin RH, Turk DC, McDermott MP, Eccleston C, Farrar JT, Rowbotham MC, Bhagwagar Z, Burke LB, Cowan P, Ellenberg SS, Evans SR, Freeman RL, Garrison LP, Iyengar S, Jadad A, Jensen MP, Junor R, Kamp C, Katz NP, Kesslak JP, Kopecky EA, Lissin D, Markman JD, Mease PJ, O'Connor AB, Patel KV, Raja SN, Sampaio C, Schoenfeld D, Singh J, Steigerwald I, Strand V, Tive LA, Tobias J, Wasan AD, Wilson HD. Interpretation of chronic pain clinical trial outcomes: IMMPACT recommended considerations. Pain. 2020 Nov;161(11):2446-2461).
The non-inferiority margin of 0.4 points on the Numeric Pain Rating Scale (NPRS) was selected based on the principle that a difference less than 50% of the active comparator’s effect compared with placebo would be an acceptable margin
- Chue P, Eerdekens M, Augustyns I, Lachaux B, Molcan P, Eriksson L, Pretorius H, David AS. Comparative efficacy and safety of long-acting risperidone and risperidone oral tablets. Eur Neuropsychopharmacol. 2005 Jan;15(1):111-7).
Data from pregabalin RCTs of at least 12 weeks’ duration demonstrated mean differences between placebo and pregabalin 300–600 mg ranging from ‒0.97 to ‒1.79 points on the NPRS, depending on the study (Reference: Lyrica, INN-Pregabalin). Based on this, a non-inferiority margin of 0.4 points was deemed appropriate.
The following text has been added: “No universally accepted clinically meaningful difference exists between treatment and comparator [23] and we considered a difference less than 50% of the active comparator’s effect compared with placebo would be an acceptable margin [24]. Data from pregabalin RCTs of at least 12 weeks’ duration demonstrated mean differences between placebo and pregabalin 300–600 mg ranging from ‒0.97 to ‒1.79 points on the NPRS, depending on the study [25]. Based on this, a non-inferiority margin of 0.4 points was deemed appropriate.”
According to Smith et al., major factors to consider in determining the clinical relevance importance of group differences go beyond statistical significant differences in the primary efficacy analysis and should include other factors such as benefit–risk profile of alternative treatments, treatment effect size compared with available treatments, safety and tolerability, rapidity of onset of action, durability of treatment effect, results on secondary endpoints, limitations of available treatments, different mechanisms of actions, cost convenience, and patient adherence, and other benefits such as drug–drug interactions. These points have been addressed in section 4 (Discussion).
Arguably, some of the results of some of the secondary outcomes are somewhat overstated.
Thank you for the comment. This will be addressed as per the minor points listed below.
Minor points
Line 60: It should be noted that ICN is only evident in a relatively small number of patients with persistent post surgical breast pain.
The reviewer is correct that intercostobrachial neuralgia (ICN) is not the only source of neuropathic pain following breast surgery. A recent review of the management of PNP following breast cancer lists the following causes: local nerve damage, most commonly to the intercostobrachial nerve (ICBN), intraoperative damage to pathways of the axillary nerve, neuroma formation, and nerve entrapment due to scar fibrosis and diminished density of intraepidermal nerve fibers in mastectomy scars, suggesting the presence of small fiber neuropathy.
- Reference: Avila F, Torres-Guzman R, Maita K, Garcia JP, De Sario GD, Borna S, Ho OA, Forte AJ. A review on the management of peripheral neuropathic pain following breast cancer. Breast Cancer (Dove Med Press). 2023 Oct 30;15:761-772.
To address this, we have updated the text follows: Intercostobrachial neuralgia (ICN), or postmastectomy pain syndrome, was initially defined by Wood in 1978 and is described as “chronic pain in the anterior aspect of the thorax, axilla, and/or upper half of the arm beginning after mastectomy or quadran-tectomy and persisting for more than three months after the surgery” [5]. This postsurgical pain is associated with nerve injury to the breast [5]. Neuropathic pain following breast surgery can be caused by local nerve damage, most commonly to the intercostobrachial nerve (ICBN), intraoperative damage to pathways of the axillary nerve, neuroma formation, and nerve entrapment due to scar fibrosis, with diminished density of intraepidermal nerve fibers in mastectomy scars suggesting the presence of small fiber neuropathy.” (Reference: Avila et al. 2023)
Line 63/64 The nomenclature of ‘post mastectomy pain syndrome’ is outdated since any breast surgery can cause the persistent pain. Indeed, most of the patients in the paper did not have a mastectomy but a lumpectomy. Approximately 1/3 patients with persistent post surgical breast pain do not have evidence of neuropathic pain and so it should be stated that this study is not necessarily applicable to all patients with persistent post-surgical breast pain.
The reviewer raises valid points regarding the outdated terminology of “postmastectomy pain syndrome” and its limited applicability to all types of breast surgery. We have revised the paragraph accordingly and no longer refer to “postmastectomy pain syndrome.”
In clinical practice, most patients following breast surgery for cancer are not routinely tested for neuropathic pain. In our opinion, there is a lack of robust data to conclusively state that only one-third of patients with persistent pain have evidence of neuropathic pain.
In our study we screened all patients for the presence of PNP with the Douleur Neuropathique en 4 Questions (DN4). Only patients with a DN4 score ≥4 were included in the sample. We are therefore confident that the data of our study is relevant for patients who experience neuropathic pain following breast cancer surgery as explicitly mentioned in the title of the manuscript.
Line 75 Ref 14 Only 9% had neuropathic pain with AI
We appreciate the comment and have removed the following text: such as aromatase inhibitors that may cause neuropathic pain.
Line 77 Is a treatment from 2009 ‘novel’?
We have replaced “novel” with “alternative”.
Line 110 Does not a positive DN4 ‘suggest’ neuropathic pain rather than ‘indicate’
We have changed “indicate” to “suggest”
Why is diabetes an exclusion criterium?
This exclusion criteria was used to streamline the patient population.
Line 118 Why are patients on higher doses opioid not eligible?
Use of high-dose opioids was excluded to streamline the patient population.
Was there a minimum NPRS for inclusion or would an NPRS of 1 be sufficient?
There was no minimum NPRS predefined. However, since only patients with present neuropathic pain according to DN4 were included, the NPRS scores in the PP population ranged from 3 to 10.
We have added the following information to section 3.2 (Participant demographics and exposure to treatment): The NPRS scores at baseline varied between 3 (min) and 10 (max).
Line 205 No detail is given of pain mapping. Was a stimulus used to map evoked pain, or was it patient indicated? If stimulus, which stimulus?
No stimulus was used; the area was determined by the patient.
Line 291 What is a zonectomy?
“Lumpectomy” is term used for removal of the tumor whereas “zonectomy” is the term used for removal of a tumor of small diameter including the zone adjacent to the tumor.
Line 299 If the maximum was two patches (line162), how can the range at 2 months be up to 4? (Not explainable by second application of 2 patches since second patch had to be 3 months after the first)
Perhaps the reviewer has overlooked the explanation in section 2.5. This section clarifies that a maximum of two patches were applied per session. If a larger area needed treatment, two sessions with an 8-day interval were scheduled. These sessions were counted as one, explaining why up to four patches could be used.
Line 334: Would the PGIC results be better expressed in a table? It is unclear if the authors took “somewhat better but the change does not make much difference” to be a ‘positive change’ or not.
We acknowledge the reviewer’s suggestion. However, as the trial was conducted within the first year of surgery, a change in Patient Global Impression of Change during this period was not anticipated. Nevertheless, we report the results, noting that nearly 1/5 of patients reported moderate but noticeable improvement with HCCP compared with 1/10 for pregabalin. These results should be interpreted cautiously due to the high number of missing data, as explained in line 348. For this reason, we prefer not to add a table for this parameter.
Line 347 For EQ5D5L, were the changes statistically different to their own baselines?
We thank the reviewer for this comment. In the HCCP arm, the change versus baseline was statistically significant (P=0.043), but not in the pregabalin arm (P=0.29). These p-values have been added to the manuscript as follows: The EQ-5D-5L utility scores in the HCCP group improved from a mean (SD) score of 0.635 (0.236) at baseline to 0.718 (0.214) at 2 months (P=0.043); in the pregabalin group, mean (SD) scores increased from 0.596 (0.23) to 0.652 (0.284) (P=0.29). These results indicate an overall enhancement in perceived health-related QoL for HCCP.
Line 355 Why were differences looked for in the different QLQ measures? This microanalysis is of dubious clinical significance.
The QLQ-C30 and BR23 are multidimensional scales and should be evaluated across all their dimensions. Please see our response to the comment on line 362.
Line 359 QOL has no statistics mentioned.
We have added the following p-values: The mean QoL score (SD) in the HCCP group increased from 56.2 (19.7) at baseline to 61.3 (20.0) at month 2 (P=0.18), compared with stable scores in the pregabalin group from 56.4 (19.4) to 56.3 (22.1) (P=0.91).
Line 362 Similarly the search for differences in the QOL dimensions seems a stretch to find significant results.
The EORTC Quality of Life Group emphasizes the importance of analyzing QoL results dimension by dimension in their guidelines and publications. Specifically, the “EORTC Quality of Life Group Manual for the Use of EORTC Measures in Daily Clinical Practice” provides detailed guidance on the presentation and interpretation of QoL data.
- Reference: The use of EORTC measures in daily clinical practice-A synopsis of a newly developed manual Lisa M Wintner 1, Monika Sztankay 2, Neil Aaronson 3, Andrew Bottomley 4, Johannes M Giesinger 5, Mogens Groenvold 6, Morten Aa Petersen 7, Lonneke van de Poll-Franse 8, Galina Velikova 9, Irma Verdonck-de Leeuw 10, Bernhard Holzner 11; EORTC Quality of Life Group Affiliations Expand, PMID: 27721057, DOI: 10.1016/j.ejca.2016.08.024
Line 369 Why are mean and medians both presented?
As the data are not normally distributed, we present both mean and median values for completeness to provide a fuller picture for the reader.
Line 382“By month 2, the mean (SD) anxiety scores decreased to 7.8 (4.1) and 8.2 (3.4), respectively, suggesting a gradual reduction in anxiety levels over the course of treatment”. This is overstating the data and no statistics are presented.
This is a difference of <1 point which corresponds to only one slightly different response to one of the 6 (graded 0-3) questions not really consistent with a ‘reduction in anxiety levels’.
The reviewer is correct as this is an interpretation of data and does not belong in the results section. Consequently, we have taken the latter part of the sentence out: By month 2, the mean (SD) anxiety scores decreased to 7.8 (4.1) and 8.2 (3.4), respectively,suggesting a gradual reduction in anxiety levels over the course of treatment.
Line 391 are there any statics associated with the tolerability data?
There are no statistics associated with the tolerability data.
Line 425 Why are these results being stated in the discussion and not the results section?
Results have been added to the results section 3.5 (QoL): After 2 months, HCCP showed important benefits in terms of cognitive capacity (P=0.006), and reduction in nausea/vomiting symptoms (P=0.023), whereas pregabalin notably enhanced work/leisure capacity (P=0.004) and future outlook (P<0.001).
P=0.05 is not significant as it is not <0.05 which is the convention.
This has been deleted.

Reviewer 3 Report
Comments and Suggestions for Authors
Post-surgical neuropathic pain is a clinically important problem affecting a small percent of patients who undergo breast surgery. Effective treatment options are limited. This paper reports a randomized clinical trial comparing local pain control with a high concentration capsaicin patch (HCCP) versus systemic therapy with oral pregabalin. The results showed that both treatments were effective and the HCCP patch was not inferior to pregabalin for reduction in pain intensity. The HCCP patch seemed to be better tolerated and preferred by the patients.
The trial was limited by small sample size due to recruitment problems related to the covid pandemic. However, the analysis was rigorous, and the results and limitations are clearly presented. HCCP is not widely used or known, at least in the United States, and this article is important in suggesting it as an alternative to pregabalin.
Author Response
Post-surgical neuropathic pain is a clinically important problem affecting a small percent of patients who undergo breast surgery. Effective treatment options are limited. This paper reports a randomized clinical trial comparing local pain control with a high concentration capsaicin patch (HCCP) versus systemic therapy with oral pregabalin. The results showed that both treatments were effective and the HCCP patch was not inferior to pregabalin for reduction in pain intensity. The HCCP patch seemed to be better tolerated and preferred by the patients.
The trial was limited by small sample size due to recruitment problems related to the covid pandemic. However, the analysis was rigorous, and the results and limitations are clearly presented. HCCP is not widely used or known, at least in the United States, and this article is important in suggesting it as an alternative to pregabalin.
We thank the reviewer for these constructive comments.

Round 2
Reviewer 1 Report
Comments and Suggestions for Authors the paper seems well written to me. I only highlight some points that need to be clarified: 1. the description of the sample size is clear, but in the text it is noted that the number actually achieved is lower than the estimated one. for this reason it would be useful to present an estimate of the power a posteriori b. Also for the sample size in the interim analysis it would be useful to have some details on how it was estimated 3. the analyzes are based on Anova for independent data (parametric method) and Wilcoxon test for paired data (non-parametric method). justify the reason for the two methods (for example the non-normal distribution or the wide dispersion in the case of paired data?)Author Response
We thank the reviewer for their careful consideration. Our responses are highlighted below in Bold, with additions to the manuscript underlined and highlighted in green. Please let us know if any further clarification or changes are needed.
Editorial comments: we considered them carefully and don’t see need for changes to the current manuscript.
Reviewer 1
Reviewer 1 comments (minor)
The paper seems well written to me. I only highlight some points that need to be clarified:
- the description of the sample size is clear, but in the text, it is noted that the number achieved is lower than the estimated one. for this reason, it would be useful to present an estimate of the power a posteriori
We thank the reviewer for the comment. A posteriori, the empirical power was calculated to be 38%, based on the allocation ratio observed in the per protocol population and with the observations from the trial assumed to be true.
We have added the following sentence to the manuscript in section 2.7.1 “If the observations in the trial are accurate and considering the allocation ratio in the per protocol population (a posteriori estimate), the empirical power is 38%.”
- Also, for the sample size in the interim analysis it would be useful to have some details on how it was estimated
We thank the reviewer for their comment, however as the interim analysis was not conducted, we consider that the level of detail already provided in section 2.7.1 is sufficient for the readers.
- the analyzes are based on Anova for independent data (parametric method) and Wilcoxon test for paired data (non-parametric method). justify the reason for the two methods (for example the non-normal distribution or the wide dispersion in the case of paired data?)
Based on the comment of the reviewer, we are providing additional detail in the manuscript to provide further clarity. In fact, when residuals followed a normal distribution, of ANOVA testing were provided. Otherwise, non-parametric tests were used as follows: the Mann-Whitney test for independent data and the Wilcoxon test for paired data.
Consequently, the relevant sentence in the manuscript in last paragraph of section 2.7.2
“Analysis of variance (ANOVA) was used to compare results between the two treat-ment groups and the Wilcoxon test was used for paired comparisons” will be changed as follows: “Results between the two treatment groups were compared with an analysis of variance (ANOVA) when residuals followed a normal distribution. Otherwise, non-parametric tests were used including the Mann-Whitney test for independent data and the Wilcoxon test for paired data.”
